# Modeling of Boring Mandrel Working Process with Vibration Damper

**DOI:** 10.3390/ma13081931

**Published:** 2020-04-20

**Authors:** Kirill Sentyakov, Jozef Peterka, Vitalii Smirnov, Pavol Bozek, Vladislav Sviatskii

**Affiliations:** 1Faculty of Technology, Votkinsk Branch of Kalashnikov Izhevsk State Technical University, 426069 Izhevsk, Russia; la1030@mail.ru (K.S.); ucheb@vfistu.ru (V.S.); svlad-2000@yandex.ru (V.S.); 2Faculty of Materials Science and Technology, Slovak University of Technology in Bratislava, Ulica Jána Bottu č. 2781/25, 917-23 Trnava, Slovakia; pavol.bozek@stuba.sk

**Keywords:** boring mandrel, vibrations, damping element, finite difference method

## Abstract

The article considers the issue of modeling the oscillations of a boring mandrel with vibration damper connected to the mandrel with a viscoelastic coupling. A mathematical model of the boring mandrel oscillations, machine support and inertial body (damper) is developed in the form of a differential equations system. The model is made in the form of a four-mass system of connected bodies. The solution to the differential equations system was found using the finite difference method, as well as the operator method with the use of the Laplace transform. As the simulation result, it was found that the use of vibration damper can significantly reduce the amplitude of the boring mandrel natural vibrations when pulsed, and also significantly reduce the forced vibrations amplitude when exposed to periodic disturbing forces. The developed mathematical model and algorithms for the numerical solution to the differential equations allowed us to choose the optimal parameters of the boring mandrel damping element. The obtained data will be used to create a prototype boring mandrel and conduct field tests.

## 1. Introduction

The most commonly used machining technologies include turning [1,2], milling [3,4] and drilling or boring [5,6]. The aim is to achieve the desired quality of the machined surface with dimensional accuracy and surface roughness [7,8]. The cutting process is characterized by the accompanying phenomena: chip formation, existence of cutting forces, tool wear [9], surface roughness [8] and vibrations. Vibration has a negative impact on the processing quality and processing performance [10]. All conventional mechanical technologies with defined cutting edge (turning, milling, drilling, boring) and so unconventional machining technologies (high speed cutting (HSC), hard machining (HM) or micro-machining) deal with the vibration phenomenon—eliminate vibrations or reduce vibrations to an acceptable level. For example, authors in [11] present an innovative method for high-speed micro-cutting of carbon fiber reinforced plastics (CFRP). The serious spindle vibration limits the rotational speed to increase further, and the rotational speed of 25,000 rpm achieves the best fine machined surface. In literature [12] was studied the prediction of chatter instability in machining steel. The chatter stability is predicted in the frequency domain using Nyquist stability criterion. In our article, we will focus on deep boring technology and vibration reduction. Solving the problem of reducing the vibration during boring is relevant for engineering production. First of all, it concerns boring mandrels for processing deep holes (the hole depth is over 5 diameters). Boring mandrel vibrations significantly reduce the processed surface quality—surface roughness and heterogeneity increase, geometric accuracy decreases. The longer the boring tool, the less rigidity it has, which means the above problems become even more acute. These difficulties force us to reduce the processing performance. The aim of the different research study is to increase the processing efficiency at boring deep holes, obtaining thus increased processing performance. In addition, it is obtained the accuracy improvement of the resulting dimensions and the quality of the machined surfaces roughness.

Methods of removing vibration during boring or drilling are technological and constructional. Technological methods solve cutting parameters optimization [9,10], selection of cutting medium (fluids) [13,14,15]. Construction methods include dynamic vibration absorbers (DVAs) [16], a new design of boring bars with different cross-sections has been considered and the use of new construction materials e.g., with new composite materials [17], construction adjustment of fiberglass sucker rod strings [18], the free vibration and chatter stability of a rotating thin-walled composite bar under the action of regenerative milling force have been investigated [19]. In paper [20], an innovative chatter suppression method based on particle damping technique is attempted to reduce chatter in a boring tool and thereby study the improvement in surface finish and tool wear. In study [21], the effect of piezoelectric shunt damping on chatter vibration in a boring process is studied. In piezoelectric shunt damping method, an electrical impedance is connected to a piezoelectric transducer which is bonded on a cutting tool. The article [22] proposes the closed-form solution for an analytical prediction of stability lobes in internal turning process. The passively damped boring bar is modeled as a cantilevered Euler–Bernoulli beam with constant cross-sectional properties, in which a tuned mass damper (TMD) is embedded for the purpose of chatter suppression. The authors in paper [23] describe the technique of particle damping in a boring process. The granular particles are packed with the single particle type used to suppress chatter. A novel flutter suppression technique is proposed using an active dynamic vibration absorber (ADVA) [24]. The ADVA is introduced by adding an active element to a classical mass-spring-damper system. In [25] a torsion-translational quasi-zero-stiffness (TT QZS) isolator with convex ball-roller mechanisms was proposed to attenuate the torsion and translational vibrations along the shaft systems simultaneously. The undesired stick-slip vibrations of the drill-string can lead to drilling failures or even to serious drilling accidents. In this paper, a new control strategy is proposed based on a state observer and a reference governor to suppress stick-slip vibrations of the drill-string [26].

According to [27], the research can be summarized and based on the different methods of vibration reduction and control of a boring bar. These studies can be divided into the following three categories: 1. The design of dynamic vibration absorbers (DVAs). 2. The design of active vibration reduction controllers. 3. To improve the dynamic stiffness of the boring bar. In the first category, various absorbers are proposed and designed—such as e.g., a passive DVA attached to boring bar [28,29], a new type of damping boring bar with a DVA [30], a three-dimensional model of a damped boring bar [31], a composite DVA with a particle damper [32], a variable-stiffness dynamic vibration absorber (VSDVA) [33]. The second category is for proposed solutions—such as e.g., the proposing a vibration reduction methodology through the use of embedded piezoelectric patches in the tool-holder [34], a 3-degree freedom linear magnetic actuator [35,36], a noncontact magnetic actuator fit with fiber optic displacement sensors, mounted on a computer numerical control (CNC) lathe [37]. In the third category, we found the following designs and solutions—such as e.g., from authors [38] four types of composite boring bars with differently shaped steel cores, the authors [39] designed and manufactured a carbon-fiber epoxy composite boring bar, the authors [40] analyzed the dynamics characteristics and stabilities of composite boring bar without considering the shear deformation and rotational inertia in an analysis.

One of the effective ways to reduce the boring mandrels vibration level is to use a dynamic vibration damper with viscous friction. The solution essence lies in the fact that the load is placed inside the boring mandrel and interacts elastically with it, for example, using rubber rings. In addition, load vibrations occur in oil, which absorbs vibrations. Today, there are some offers of boring steel or carbide mandrels with dynamic vibration dampers from cutting tools manufacturers Sandvik Coromant (Sandviken, Sweden), WIDIA Production Group (Furth, Germany) or IMC Group ISCAR (Tel Aviv-Yafo, Israel). These are the solutions such as Silent Tools™, Whisperline and others. Anti-vibration mandrels with dynamic vibration dampers are typically used for boring holes up to 14 × D in depth. The claimed increase in productivity when using such mandrels is from 50% to 400%, depending on the range of depth/diameter ratios. Some of the existing solutions have the ability to configure the mandrel for a specific situation with the selected cutting conditions and the workpiece parameters. This allows to optimize the damping characteristics for each specific case and reduce the range of tools.

The basis of the model is the classical system of differential equations, which describes the balance of forces in the system according to the famous Newton’s law. A new approach is proposed to solve such a system by presenting it in the form of a structural diagram of dynamic input-output blocks. This makes it easier to carry out simulation in specialized application programs. Using graph theory methods (Mason’s method) in a structural diagram, it is possible to compose the necessary Transfer Functions of any impact-displacement pair, that is, to reduce Equation (1) to one differential equation with respect to any unknown. In addition, by transfer functions, simply converting them into frequency transfer functions, obtain the amplitude-phase frequency characteristics (Nyquist diagram) and conduct frequency analysis for any link in the dynamic system.

## 2. Design of Mathematical Model

During observations of the boring process, we found that it is advisable to include the following bodies in the design scheme:machine support with a tool holder;boring mandrel;inertial body (damper).

The characteristics of the listed bodies (mass, fixing rigidity or damping coefficient) significantly affect the oscillations of the tool cutting part. Figure 1 shows the machine node for which the calculation is made. General theoretical principles for such vibrational systems study have been considered in works [41,42,43,44], on the basis of which this study has been based.

We created the mathematical model of the system and we consider it as the system of interacting bodies connected by a viscoelastic coupling. In this case, the mandrel is divided into 2 parts: a less rigid hollow part, in which the damper is placed, and a more rigid part. We consider the system oscillations under the action of the variable radial component of the cutting force *P*_y_, which, for simplicity, will be denoted by *P*. The deformation in the direction of the force *P*_y_ is the most critical from the point of the view of item size accuracy, waviness and surface roughness. The designed scheme of the system is presented in Figure 2. Figure 2 presents the parallel effect of masses *m*_2_ and *m*_3_ on mass *m*_1_.

In Figure 2: *m* is the element mass (or equivalent given mass), kg; *c* is the connection elements stiffness (or the bending stiffness of the boring mandrel), N/m; *a* is the damping coefficient that determines the energy loss due to viscous friction, kg/s; *P(t)* is the disturbing force from the cutting process, N; *V(t)* is the machine vibration transmitted to the machine support with a tool holder, m; *y* is the movement of elements from the static equilibrium position, m.

Indexes relate to the following elements: 1: flexible (hollow) part of the cutter; 2: damper; 3: the hard part of the cutter; 4: tool holder with machine support.

Of greatest interest in the calculation is *y*_1_—the movement of the mandrel flexible part (the movement of the cutting part).

A variable cutting force *P* acts on the boring mandrel, which tends to deflect it from the static equilibrium position. Measurements of boring mandrel oscillations showed that the prevailing vibration frequencies are:the natural vibrations frequency (approximately 150–250 Hz) that arise due to the occurrence of self-oscillations characteristic of the sharpening process;the forced vibrations frequency (approximately 400–600 Hz) that arises as a result of the chip formation process.

The average value of *P* depends on the applied cutting conditions.

The experiments also showed that the mandrel oscillation frequencies are scattered, even under constant cutting conditions. It is important that the designed vibration damper had a sufficiently wide band of an effective vibration absorption.

Next, we composed a mathematical model. Based on Newton’s second law, we consider the balance of forces acting on each of the four masses. In this model, two forces are considered:the elastic resistance force, proportional to the movement,the viscous resistance force, proportional to the speed of movement.

Four equations of forces equilibrium make up a system of four differential equations in accordance with the calculation scheme in Figure 2:
(1)$$\{\begin{array}{l} m_{1}y_{1}^{\prime \prime }=P+c_{1}\left(y_{3}-y_{1}\right)+c_{2}\left(y_{2}-y_{1}\right)+a_{1}\left(y_{3}^{\prime }-y_{1}^{\prime }\right)+a_{2}\left(y_{2}^{\prime }-y_{1}^{\prime }\right)\\ m_{2}y_{2}^{\prime \prime }=c_{2}\left(y_{1}-y_{2}\right)+a_{2}\left(y_{1}^{\prime }-y_{2}^{\prime }\right)\\ m_{3}y_{3}^{\prime \prime }=c_{1}\left(y_{1}-y_{3}\right)+c_{3}\left(y_{4}-y_{3}\right)+a_{1}\left(y_{1}^{\prime }-y_{3}^{\prime }\right)+a_{3}\left(y_{4}^{\prime }-y_{3}^{\prime }\right)\\ m_{4}y_{4}^{\prime \prime }=c_{3}\left(y_{3}-y_{4}\right)+c_{4}\left(V-y_{4}\right)+a_{3}\left(y_{3}^{\prime }-y_{4}^{\prime }\right)+a_{4}\left(V^{\prime \prime }-y_{1}^{\prime }\right) \end{array}$$

The free movement of each mass can be individually described by a second-order differential equation:(2)$$m_{i}y_{i}^{\prime \prime }+a_{i}y_{i}^{\prime }+c_{i}y_{i}=0$$

Or in the form of a typical oscillatory link [45] of a dynamic system:(3)$$T_{i}^{2}y_{i}^{\prime \prime }+2T_{i}\xi_{i}y_{i}^{\prime }+y_{i}=0$$
where *Ti* is the time constant or free oscillations period, s; *ξi* is the damping coefficient (0 < *ξ_i_* < 1). These parameters, when jointly considering Equations (2) and (3) make the following:(4)$$T_{i}=\sqrt{\frac{m_{i}}{c_{i}}}$$
(5)$$a_{i}=2T_{i}\xi_{i}c_{i}$$

Then the natural frequency of each of the four elements is determined by the well-known [45] formula:(6)$$\omega_{i}=\frac{\sqrt{1-\xi_{i}^{2}}}{T_{i}}$$

To study the object in the frequency domain and also to consider the alternative ways of the solving Equation (1), it is proposed to use the methods of [45,46,47] Automatic Control Theory (TAU). By applying Laplace transforms and transition to the representation of the Equation (7):(7)$$\{\begin{array}{l} y^{\prime \prime }\to p^{2}Y_{i}\\ y_{i}^{\prime }\to pY_{i} \end{array}$$
the Equation (1) can be represented in the Equation (8):(8)$$\{\begin{array}{l} m_{1}p^{2}Y_{1}+\left(a_{1}+a_{2}\right)\;pY_{1}+\left(c_{1}+c_{2}\right)\;Y_{1}=P+\left(c_{2}+a_{2}p\right)\;Y_{2}+\left(c_{1}+a_{1}p\right)Y_{3}\\ m_{2}p^{2}Y_{2}+a_{2}pY_{2}+c_{2}Y_{2}=\left(c_{2}+a_{2}p\right)Y_{1}\\ m_{3}p^{2}Y_{3}+\left(a_{1}+a_{2}\right)\;pY_{3}+\left(c_{1}+c_{3}\right)\;Y_{3}=\left(c_{1}+a_{1}p\right)\;Y_{1}+\left(c_{3}+a_{3}p\right)Y_{4}\\ m_{4}p^{2}Y_{4}+\left(a_{3}+a_{4}\right)\;pY_{4}+\left(c_{3}+a_{3}p\right)\;Y_{3}+\left(c_{4}+a_{4}p\right)V \end{array}$$

Or as an operator equations system with transfer functions:(9)$$\{\begin{array}{l} Y_{1}=W_{11}P+W_{12}Y_{2}+W_{13}Y_{3}\\ Y_{2}=W_{2}Y_{1}\\ Y_{3}=W_{31}Y_{1}+W_{32}Y_{4}\\ Y_{4}=W_{41}Y_{3}+W_{42}V \end{array}$$
where the operator transfer functions are defined as follows in Equation (10) and they are the mathematical models of dynamic system structural elements (Figure 3).
(10)$$\{\begin{array}{l} W_{11}=\frac{1}{m_{1}p^{2}+\left(a_{1}+a_{2}\right)p+\left(c_{1}+c_{2}\right)}\\ W_{12}=\frac{c_{2}+a_{2}p}{m_{1}p^{2}+\left(a_{1}+a_{2}\right)p+\left(c_{1}+c_{2}\right)}\\ W_{13}=\frac{c_{1}+a_{1}p}{m_{1}p^{2}+\left(a_{1}+a_{2}\right)p+\left(c_{1}+c_{2}\right)}\\ W_{2}=\frac{c_{2}+a_{2}p}{m_{1}p^{2}+a_{2}p+c_{2}}\\ W_{31}=\frac{c_{1}+a_{1}p}{m_{3}p^{2}+\left(a_{1}+a_{2}\right)p+\left(c_{1}+c_{2}\right)}\\ W_{32}=\frac{c_{3}+a_{3}p}{m_{3}p^{2}+\left(a_{1}+a_{2}\right)p+\left(c_{1}+c_{2}\right)}\\ W_{41}=\frac{c_{3}+a_{3}p}{m_{4}p^{2}+\left(a_{3}+a_{4}\right)p+\left(c_{3}+c_{4}\right)}\\ W_{42}=\frac{c_{4}+a_{4}p}{m_{4}p^{2}+\left(a_{3}+a_{4}\right)p+\left(c_{3}+c_{4}\right)} \end{array}$$

Then, solving Equation (9) with respect to *Y*_1_, obtains the general transfer Equation (11) for the force *P* and disturbance *V*, that is, the solution of Equation (1) in an operator form.
(11)$$\{\begin{array}{l} Y_{1}=W_{p}P+W_{V}V\\ W_{p}\left(p\right)=\frac{W_{11}\left(1-W_{41}W_{32}\right)}{1-W_{2}W_{12}-W_{31}W_{13}-W_{41}W_{32}+W_{2}W_{12}W_{41}W_{32}}\\ W_{p}\left(p\right)=\frac{W_{11}\left(1-W_{41}W_{32}\right)}{1-W_{2}W_{12}-W_{31}W_{13}-W_{41}W_{32}+W_{2}W_{12}W_{41}W_{32}} \end{array}$$

By performing the inverse Laplace transform, it is possible to obtain an analytical solution to the task. Passing to the complex transfer Equation (12), according to the well-known formulas [45], we have the amplitude-frequency *A*(*ω*) and phase-frequency *φ*(*ω*) characteristics:(12)$$\{\begin{array}{l} W_{p}\left(\omega i\right)=Re\left(\omega \right)+Im\;\left(\omega \right)i\\ A\left(\omega \right)=\left|W_{p}\left(\omega i\right)\right|=\sqrt{Re{\left(\omega \right)}^{2}+Im{\left(\omega \right)}^{2}}\\ \varphi \left(\omega \right)=\arg\left(W_{p}\left(\omega i\right)\right)=arctg\left(Im\left(\omega \right)/Re\left(\omega \right)\right) \end{array}$$

## 3. Results

System analytical solution (1) leads to a single linear differential equation with constant eighth-order coefficients. A further solution involves finding the roots of the eighth-degree characteristic equation and considering various solutions for real or complex roots, which causes certain difficulties. Therefore, Equation (1) was solved by a numerical iterative method of finite differences. Figure 4 shows the solution with a stepwise action of the force *P* = 500 N and in the absence of disturbance *V*. Modeling was carried out with the following system parameters: *m*_1_ = 2 kg, *m*_2_ = 2 kg, *m*_3_ = 6 kg, *m*_4_ = 20 kg, *c*_1_ = 6 × 106 N/m, *c*_2_ = 2 × 106 N/m, *c*_3_ = 12 × 106 N/m, *c*_4_ = 25 × 106 N/m, *a*_1_ = 50 kg/s, *a*_2_ = 1000 kg/s, *a*_3_ = 50 kg/s, *a*_4_ = 100 kg/s.

To assess the vibration damper effectiveness, the response of the tool holder to the pulsed action is compared with and without damper. Figure 5 presents a comparison of the impulse responses (*y*_1_) for a single impulse action of a force *P* = 50 kN with a duration of 0.00002 s (the rectangular impulse integral is 50,000 × 0.00002 = 1). Figure 6 presents a comparison of the working process under the frequency action of the force with an amplitude of *P* = 500 N at a frequency of *ω* = 900 rad/s (≈ 140 Hz). The comparison data clearly confirm the vibration damper effectiveness in the tool holder.

Figure 7 shows the amplitude-frequency characteristics (AFC) for mass 1 with a damper and without a damper under the frequency action of a force *P* = 500 N, with the same parameters as for the numerical system solution (1) in Figure 3. This dependence allows estimating the range of dynamic system working frequencies, as well as its critical frequencies at which resonance phenomena occur.

The critical frequencies themselves can be calculated without the frequency response, as the square roots of the matrix eigenvalues (13) of the coefficients of the Equation (1) non-damped (*ξi* = 0, *ai* = 0) system of differential equations.

The analytical calculation of the eigenvalues of the fourth-order matrix again requires high-order algebraic equations solution. However, there are a lot of applied software solutions for such problems, with the help of which four critical frequencies were found for the dynamic system parameters adopted above: 2358, 1653, 665 and 1055 rad/s. The solution was obtained in Math Studio. Simulation was carried out in VisSim version 3.0E. You can use the Mathlab Simulink package.
(13)$$\left[\begin{array}{llll} \frac{-\left(c_{1}-c_{2}\right)}{m_{1}} & \frac{c_{2}}{m_{1}} & \frac{c_{1}}{m_{1}} & 0\\ \frac{c_{2}}{m_{2}} & -\frac{c_{2}}{m_{2}} & 0 & 0\\ \frac{c_{1}}{m_{3}} & 0 & \frac{-\left(c_{1}+c_{3}\right)}{m_{3}} & \frac{c_{2}}{m_{3}}\\ 0 & 0 & \frac{c_{3}}{m_{4}} & \frac{-\left(c_{3}+c_{4}\right)}{m_{4}} \end{array}\right]$$

For the disturbing vibrational effect *V*, in a similar way, from the second transfer Equation (11), we can obtain one’s own frequency response, which will have the same shape, the same critical frequencies, but different values along the ordinate. This will allow to evaluate the component of machine vibration influence on the considered dynamic system. As shown by practical experience in the use of such tool holders, this issue is very relevant.

In addition, solving Equation (9) with respect to any other *Y_i_*, one can study the dynamics and frequency characteristics of any of the four masses participating in the calculation model (Figure 2).

## 4. Discussion

In the course of the damping process research, connections have been shown to make the integration of the scheme of the following three attributes conditional: support of production equipment (weight, fastening rigidity or damping coefficient) have a significant influence on the process of selecting the cutting conditions of drilling tool. The theoretical basis without further research is described in the aforementioned scientific studies [41,42,43,44]. They did not specify model parameters for vibration amplitude during the research. In the following discussion, we present a simulation of a mandrel with a dynamic vibration damper with concrete results.

Thus, the simulation showed that the use of a mandrel with a dynamic vibration damper in the described design leads to an increase in the processing efficiency by boring deep holes. In particular, for the accepted model parameters, the vibration amplitude during a pulse action on the tool could be reduced by 2–3 times from 600 μm to 200 μm, and the transition time is reduced from 0.2 s to 0.05 s (Figure 5). The steady-state vibration amplitude of the instrument during a frequency exposure from 400 microns to 100 microns also significantly decreased (Figure 6). The resulting mathematical model made it possible to determine the critical frequencies of the instrumental system without damping and with damping, which for the adopted model parameters are 2358, 1653, 665 and 1055 rad/s (Figure 7). The model confirmed the possibility of expanding the frequency range of the resonant-free operation of the tool holder by effectively suppressing its critical frequencies with a dynamic vibration damper. The model can be used in the design of mandrels with a dynamic vibration damper for selecting their parameters for a given range of operating frequencies.

In the future, by developing the application of TAU methods in this matter, it is possible to study the dynamic system stability analytically, for example, according to the Hurwitz criterion, with stability areas construction according to certain parameters. It is possible to optimize the system parameters to obtain the best transient response, for example, by integral estimates, using computer simulation, artificial intelligence [48], using multi-criteria analysis [49], prediction of a new form of the cutting tool on base of machinability tests [50], using Petri nets for modeling and performance evaluation of discrete event systems [51], using a numerical analysis of the stress-strain [52] or other. The numerical solution to the differential Equation (1) can be used for simulation of the tool holder working process in various modes for various parameters, various new materials too [53]. It can be stated that the proposed device and the mathematical model contribute to increasing the stability of the boring process, to achieving greater accuracy and quality of machined surfaces. This is in accordance with the logic “the quality of technological machines and equipment is related to the quality of products” [54,55].

On the base of the achieved results, some generic and fundamental academic conclusions is possible to be drawn. First of all, the main meaning of reduction in vibration amplitude by design results in technological optimization of the deep drilling process—boring, divided into the following three steps:

A. If the vibrations are reduced then it is possible to adequately increase the technological parameter (cutting speed) while maintaining the tool life, but if the cutting speed increases (by the increase of rotation number of tool) then the feed boring speed of the deep boring process increases too and thus the productivity of the deep boring process increases.

B. If the vibrations are reduced then it is possible to adequately increase the technological parameter (feed per tooth) while maintaining the good roughness of machined surface (boring hole surfaces), but if the feed per tooth increases then the feed boring speed of the deep boring process increases too and thus the productivity of the deep boring process increases.

C. If the vibrations are reduced then it is possible to adequately increase the both technological parameters (cutting speed and feed per tooth) while maintaining the tool life and the good roughness of machined surface (boring hole surfaces), but if the cutting speed increases (by the increase of rotation number of tool) and simultaneously the feed per tooth increases then the feed boring speed of the deep boring process increases too and thus the productivity of the deep boring process increases.

## 5. Conclusions

A dynamic model of a boring mandrel with a machine tool system was presented. The progress in understanding of knowledge presented in the work is as follows:
This dynamic model was made including the mandrel separation into segments with different parameters of:-mass,-stiffness and damping.The segmentation allowed to describe the operation of the hole boring process with such a mandrel with a damper more accurately.The proposed representation of a mathematical model of differential equations system in the form of a structural diagram of interacting dynamic elements with transfer functions allowed:-to obtain a common frequency complex transfer function of the entire system,-for its subsequent transformation into an amplitude-frequency characteristic.This solution allows to find a safe frequency range with a minimum amplitude of the instrumental system vibration.Important for the technological process for this dynamic mathematical model of the technological system is the finding that the vibration amplitude:-can be reduced 2–3 times during impulse action on the tool,-at steady state during a frequency exposure also decreases significantly up to 4 times.

In the future, it is planned to conduct a number of field experiments to assess the adequacy of the proposed mathematical model, and its further correction. It can be either experiments on real equipment and rigging directly at the Votkinsky Zavod in Izhevsk, Russa or bench tests in the technological laboratory of Centre of Excellence of 5-Axis Machining at the Slovak University of Technology in Trnava or in the laboratories of the Votkinsk branch of the Izhevsk State Technical University named after M.T. Kalashnikov, Russia.

## Figures and Tables

**Figure 1 materials-13-01931-f001:**
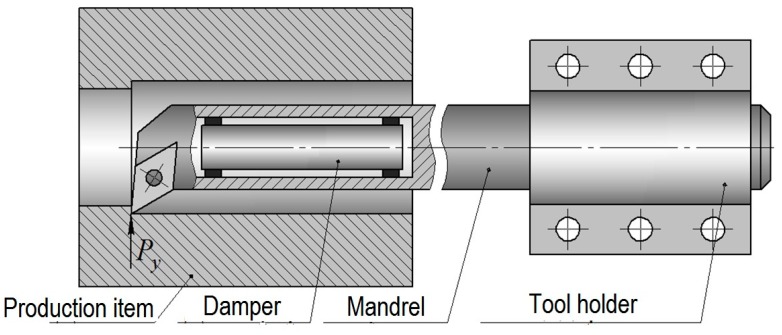
The machine node (schematically).

**Figure 2 materials-13-01931-f002:**
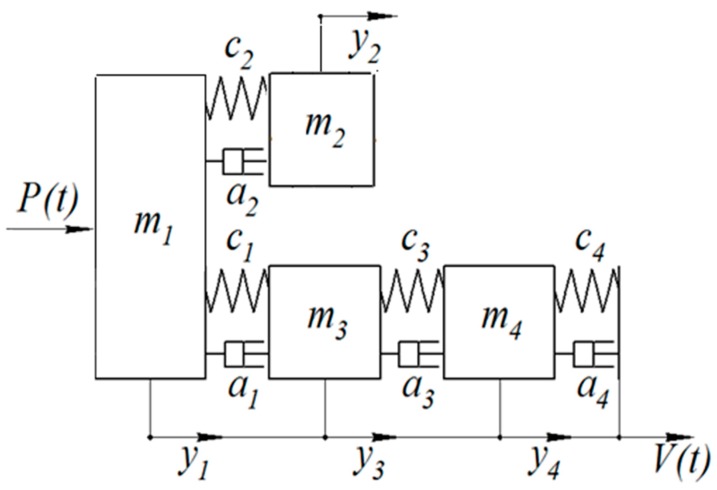
The designed scheme.

**Figure 3 materials-13-01931-f003:**
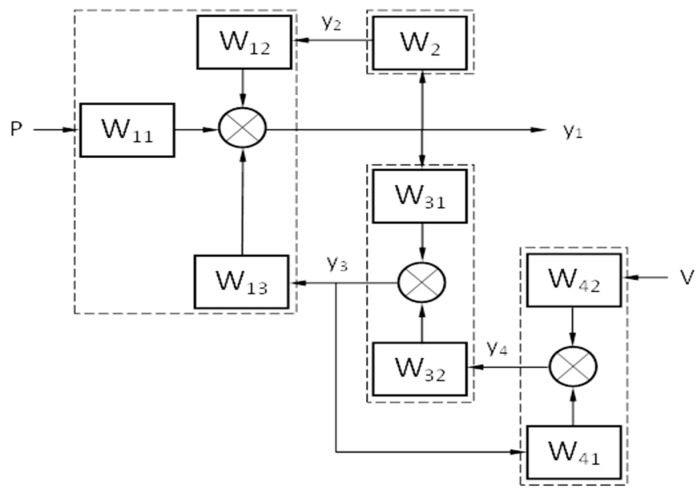
Structural model of dynamic system.

**Figure 4 materials-13-01931-f004:**
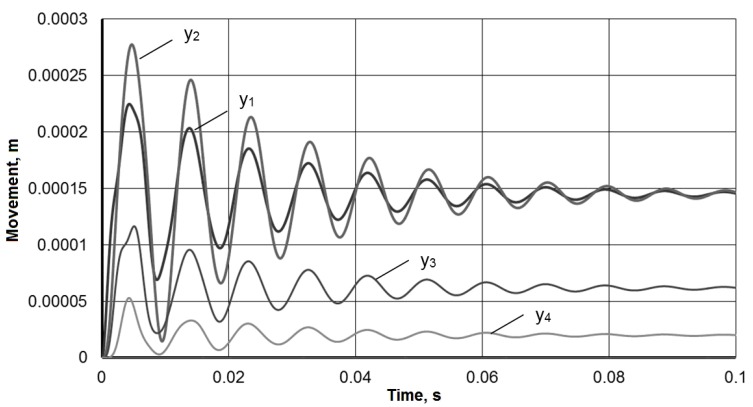
System numerical solution in Equation (1).

**Figure 5 materials-13-01931-f005:**
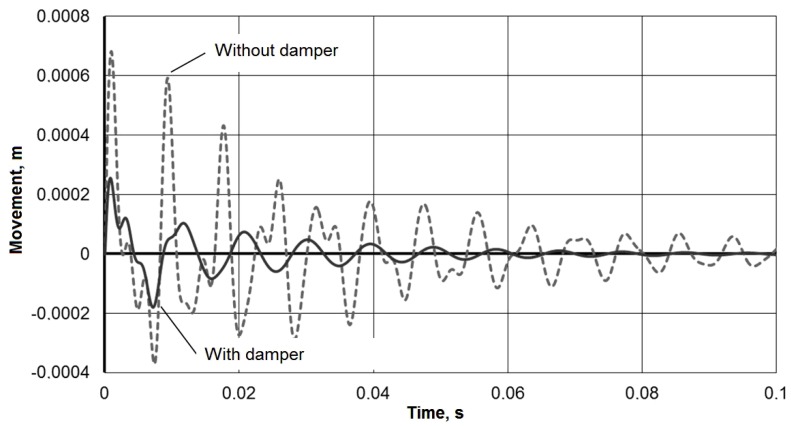
Impulse responses comparison.

**Figure 6 materials-13-01931-f006:**
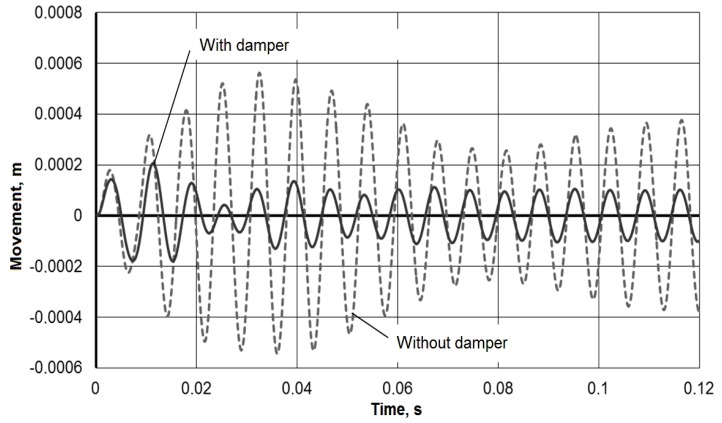
Frequency modes comparison.

**Figure 7 materials-13-01931-f007:**
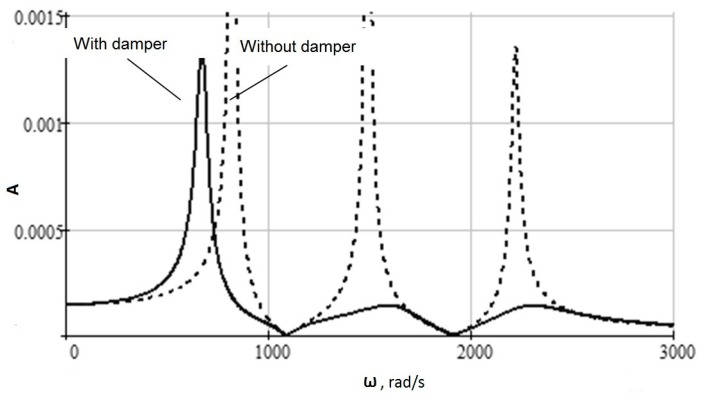
Frequency response.

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
