# Peer review of "Modeling of Boring Mandrel Working Process with Vibration Damper"

_materials, 2020, doi:10.3390/ma13081931_

Round 1
Reviewer 1 Report
This paper introduces mathematical analysis of boring mandrel with a vibration damper inside the mandrel. Detailed mathematical model and simulation results are discussed. This paper is suitable to the Special Issue "Precision and Ultra-Precision Subtractive and Additive Manufacturing Processes of Alloys and Steels". There are following problems.
• Structure of the paper needs to be improved. There are too many paragraphs in some sections.
• Some numerical research on the boring mandrel system, if possible, can be added to the introduction section.
• There are some writing and grammar errors in the paper.
• According to Fig. 1, the damper and the flexible (hollow) part of the cutter are of parallel relationship. Why do you build the math model, shown in Fig. 2, in a serial form?
• Surface roughness has not been discussed in the article in detail, while ‘surface roughness’ is a keyword. The same question is also to the keyword ‘hole accuracy’.
• The conclusions are too long and some part can be moved to discussion.
• It seems that there is a relationship among stiffness values (c1, c2, c3 and c4), what is the relationship of their values in the model? Why do you set these values?
• A comparison between the give boring mandrel structure and commonly used structure can be added.
• Materials of the mandrel are not introduced, and properties of these materials are not discussed.
• Rotation of the cutter is not considered in the model, thus bending and moment on the mandrel are not concerned. Does this ignorance affect model accuracy?
• If possible, experimental validation/comparison can be added.
Author Response
Point 1: Structure of the paper needs to be improved. There are too many paragraphs in some sections.
Response 1: The number and size of paragraphs significantly affects the clarity of the article. We agree with the opponent that there are too many paragraphs in our article and that is why we have merged the following paragraphs:
the original Line 33-48 to the current Line 33-55 including the new inserted text (blue color),
the original Line 49-74 to the current Line 56-79,
the original Line 75-87 to the current Line 96-107,
the original Line 93-95 and 98-99 to the current Line 124-127,
the original Line 216-229 to the current Line 254-265,
the original Line 230-242 to the current Line 266-278.
Point 2: Some numerical research on the boring mandrel system, if possible, can be added to the introduction section.
Response 2: We agree with the opponent that it would be appropriate to introduce numerical research on the boring mandrel systems in the introduction. We believe that the introduction provides an overview of the studies. We think our approach is unique and that is why it is difficult to find the same research. We appreciate this comment, so we added in the beginning of the article some several other current literary sources [27-40] very close to our research. New inserted literatures are colored (shaded) yellow. New inserted paragraph is colored (shaded) yellow too. The current Line 80-95:
“According to [27], the research can be summarized on the based on the different methods of vibration reduction and control of a boring bar, these studies can be divided into the following three categories: 1. The design of dynamic vibration absorbers (DVAs). 2. The design of active vibration reduction controllers. 3. To improve the dynamic stiffness of the boring bar. In the first category various absorbers are proposed and designed – such as e.g. a passive DVA attached to boring bar [28,29], a new type of damping boring bar with a DVA [30], a three-dimensional model of a damped boring bar [31], a composite DVA with a particle damper [32], a variable-stiffness dynamic vibration absorber (VSDVA) [33]. For second category are proposed solutions – such as e.g. the effect of piezoelectric shunt damping on the chatter vibrations in a boring process [34], a 3-degree freedom linear magnetic actuator [35,36], a noncontact magnetic actuator fit with fiber optic displacement sensors, mounted on a computer numerical control (CNC) lathe [37]. In the third category we can found following designs and solutions – such as e.g. from authors [38] four types of composite boring bars with differently shaped steel cores, the authors [39] designed and manufactured a carbon-fiber epoxy composite boring bar, the authors [40] analyzed the dynamics characteristics and stabilities of composite boring bar without considering the shear deformation and rotational inertia in an analysis.”
Point 3: There are some writing and grammar errors in the paper.
Response 3: The some writing and grammar errors were found. The revisions of these errors were clearly highlighted using system: Word-System Review-New Comment.
Grammar errors:
- The text was cleaned of grammar errors.
Writing errors that we found:
- The original Line 68: 2x “a classical”, the current Line 74: One word “a classical” has been deleted.
- The original Line 151: “Applying the Laplace transforms, and going to the images (7), system (1) can be represented in the form (8):” - unreadable wording.
The current Line 180-181: “By applying Laplace transforms and transition to the representation of the equation system (7), the system (1) can be represented in the form (8):” - corrected wording.
- The original Line 105: "will be denoted by R" - wrong symbol R. The current Line 134: "will be denoted by P" – corrected symbol P.
Point 4: According to Fig. 1, the damper and the flexible (hollow) part of the cutter are of parallel relationship. Why do you build the math model, shown in Fig. 2, in a serial form?
Response 4: We would like to provide the following explanation to this comment: Perhaps pic. 2 is not quite correct. The mathematical model (1) and the structural model (Fig. 3) describe these two elements in parallel connection.
Point 5: Surface roughness has not been discussed in the article in detail, while ‘surface roughness’ is a keyword. The same question is also to the keyword ‘hole accuracy’.
Response 5: Yes, we agree with the opponent that the article does not primarily deal with the research of the roughness of machined surfaces and the accuracy of the dimensions of the drilled holes. These terms are mentioned in our article because tool vibrations have a significant effect on these parameters. In conclusion, we state that the proposed model, we plan to build experimentally and test on a real drilling process. Therefore, based on the opponent's reminder, we removed the words roughness and accuracy from the keywords.
Point 6: The conclusions are too long and some part can be moved to discussion.
Response 6: Yes, we agree with the opponent's comment. We moved the original lines 270-287 to the discussion chapter, now they are the current lines 279-296.
At the same time, we deleted the original lines 264-269 point 6 for the comments of the next opponent, but also for your previous comments on the "roughness" and "accuracy" keywords.
Point 7: It seems that there is a relationship among stiffness values (c1, c2, c3 and c4), what is the relationship of their values in the model? Why do you set these values?
Response 7: Thank you for your question. Our answer is as follows we hope that our explanation will be sufficient. In the model, the stiffnesses of the elements are independent quantities and, together with the masses, determine the natural frequencies of the elements.
Point 8: A comparison between the give boring mandrel structure and commonly used structure can be added.
Response 8: Thank you for your feedback. Yes, it is possible. Perhaps this description of Fig. 1 and Fig. 2.
Point 9: Materials of the mandrel are not introduced, and properties of these materials are not discussed.
Response 9: Thank you for your question. Our answer is as follows: The materials of the components of the mandrel are determined through their masses and elastic properties, i.e. coefficients (c1, c2, c3 and c4).
Point 10: Rotation of the cutter is not considered in the model, thus bending and moment on the mandrel are not concerned. Does this ignorance affect model accuracy?
Response 10: Very good question. There are technological cases where the drilling tool rotates – e.g. classic drilling on a drilling machine, or the drilling tool does not rotate – e.g. the drilling on a turning machine. Then consider turning boring in which the cutter does not really rotate.
Point 11: If possible, experimental validation/comparison can be added.
Response 11: Thank you very much for the question or comment. It is very interesting and will be carried out in further studies.
Reviewer 2 Report
- The equivalent physical quantity in Figure 2 corresponds to which component name should be clearly marked.
- The optimal parameters of the boring mandrel damping element mentioned in Line 28 is a good idea. If there is a variable damping system, it will help improve the accuracy of boring.
- This research constructs a vibration damping device in the bored mandrel, which can effectively reduce vibration during processing. In terms of improving processing quality, such as: improving surface roughness, shape accuracy, and material removal rate. Is there quantitative data to show the improvement rate?
- In the future, this numerical analysis mode will be used to verify the accuracy of the model estimation through experiments. What correction factors can be added to the model for correction?
- Should the resonance boundary conditions generated during boring be considered in the numerical analysis model?
Author Response
Point 1: The equivalent physical quantity in Figure 2 corresponds to which component name should be clearly marked.
Response 1: Thank you for your comment, we would like to explain that it means that during bending movements, the moment of inertia will be the equivalent value.
Point 2: The optimal parameters of the boring mandrel damping element mentioned in Line 28 is a good idea. If there is a variable damping system, it will help improve the accuracy of boring.
Response 2: Agree. (The original Line 28, the current Line 28)
Point 3: This research constructs a vibration damping device in the bored mandrel, which can effectively reduce vibration during processing. In terms of improving processing quality, such as: improving surface roughness, shape accuracy, and material removal rate. Is there quantitative data to show the improvement rate?
Response 3: Thank you for a very good question. Experimental studies are certainly planned in future work.
Point 4: In the future, this numerical analysis mode will be used to verify the accuracy of the model estimation through experiments. What correction factors can be added to the model for correction?
Response 4: Most likely, you will have to introduce a correction factor for the damping index in the original equation (3) or determine it experimentally.
Point 5: Should the resonance boundary conditions generated during boring be considered in the numerical analysis model?
Response 5: Thank you for your question. Let us explain. The model allows you to analytically determine the resonant frequencies that should be taken into account and avoided when choosing processing modes for boring.
Reviewer 3 Report
· Introduction (lines34-39) : too short. More information and discussion is needed on different technologies (conventional, non-traditional machining, etc) to provide better context for this study.
· Line 68-69: rephrase (2x “classical)”
· Introduction: not clear why the study is needed and what is exactly the novelty of this study! Add the objectives of the study together with its needs (such model does not exist yet? Why/how is this one better?!
· Line 101-102: improve introduction of this section 2 M and M.
· Line 151-15: improve readability and wording.
· Section 3, lines 175-178: why these values? Please explain and provide context!
· Line 201: which software solution is applied here?
· Section 4: Please improve introduction of this section!
· Conclusion: it is not clear why and how this sudy resulted in the mentioned conclusions (e.g. how can you proof conclusion (6) with the conducted study??)
Author Response
Point 1: Introduction (lines34-39) : too short. More information and discussion is needed on different technologies (conventional, non-traditional machining, etc) to provide better context for this study.
Response 1: Yes, we agree with the opponent's view that the introductory part in the introduction is too short. To clarify and improve input quality, we have expanded this section with the following text: The current Line 38-47. The inserted text is colored (shaded) blue:
“All conventional mechanical technologies with defined cutting edge (turning, milling, drilling, boring) and so unconventional machining technologies (high speed cutting (HSC), hard machining (HM) or micro-machining) deal with the vibration phenomenon - eliminate vibrations or reduce vibrations to an acceptable level. An example authors in [11] present an innovative method for high-speed micro-cutting of carbon fiber reinforced plastics (CFRP). The serious spindle vibration limits the rotational speed to increase further, and the rotational speed of 25,000 rpm achieves the best fine machined surface. In literature [12] was study the prediction of chatter instability in machining steel. The chatter stability is predicted in the frequency domain using Nyquist stability criterion. In our article, we will focus on deep boring technology and vibration reduction.”
So, we inserted in chapter References the current literary sources [11-12] cited in this paragraph.
At the same time, we point out that another opponent had a reminder that there are many paragraphs at work. In this context, we have also merged the above the original Line 34-48 entry.
Point 2: Line 68-69: rephrase (2x “classical)”
Response 2: Yes, this is a writing error. Thank you very much for this reminder. One word "classic" has been deleted. The current Line 74.
Point 3: Introduction: not clear why the study is needed and what is exactly the novelty of this study! Add the objectives of the study together with its needs (such model does not exist yet? Why/how is this one better?!
Response 3: Yes, we agree with the opponent and let us include the following text in the introduction.
The current Line 108-117. The inserted paragraph is colored (blue) blue:
“The basis of the model is the classical system of differential equations, which describes the balance of forces in the system according to the famous Newton's law. But a new approach is proposed to solve such a system by presenting it in the form of a structural diagram of dynamic input-output blocks. This makes it easier to carry out simulation in specialized application programs. Using graph theory methods (Mason's method) in a structural diagram, it is possible to compose the necessary Transfer Functions of any impact-displacement pair, that is, to reduce system (1) to one differential equation with respect to any unknown. Also, by transfer functions, simply converting them into frequency transfer functions, obtain the amplitude-phase frequency characteristics (Nyquist diagram) and conduct frequency analysis for any link in the dynamic system.“
Point 4: Line 101-102: improve introduction of this section 2 M and M.
Response 4: Yes, we can agree with the opponent and thank you for reminding us that the original title of Chapter 2. Materials and Methods and the entry into this chapter were somewhat unclear and did not describe the content of the chapter. Therefore, we have clarified the title of Chapter 2. and from the original title “2. Materials and Methods” to the current title “2. Design of Mathematical Model”.
In addition, we think that the original Lines 88-99 and the Figure 1 are an appropriate input to this chapter and therefore we have included and moved them in Chapter 2 as input (the current Lines 119-129).
Point 5: Line 151-15: improve readability and wording.
Response 5: We assume that the opponent meant the sentence in the original lines of Line 151-152. We have reworded this sentence (the current lines are Line 180-181).
The origin sentence: “Applying the Laplace transforms, and going to the images (7), system (1) can be represented in the form (8):”
The reworded sentence: “By applying Laplace transforms and transition to the representation of the equation system (7), the system (1) can be represented in the form (8):”.
Point 6: Section 3, lines 175-178: why these values? Please explain and provide context!
Response 6: The current lines 204-207. Thank you for this good question and would be happy to respond as follows: The values are selected in accordance with the expert assessment of a specialist in a machine-building enterprise and approximately describe the real technological system of turning boring.
Point 7: Line 201: which software solution is applied here?
Response 7: Thera are a lot of applied software solution. We will answer your question as follows:
The solution was obtained in Math Studio: http://mathstud.io. Simulation was carried out in VisSim: https://web.solidthinking.com/vissim-is-now-solidthinking-embed - version 3.0E is a free academic version. You can use the Mathlab Simulink package: https://www.mathworks.com/products/simulink.html
At the same time, we inserted this text into the article text, the current Line 232-233. The inserted text is colored (shade) blue.
“The solution was obtained in Math Studio. Simulation was carried out in VisSim. You can use the Mathlab Simulink package.”
Point 8: Section 4: Please improve introduction of this section!
Response 8: Thank you very much for your comment. To improve the introduction of this section, we have included the following text, which should appropriately include this chapter. The current Line 246-253. The inserted paragraph is colored (shaded) blue:
“In the course of the damping process research, connections have been shown to make the integration of the scheme of the following three attributes conditional: support of production equipment with tool holder, drill mandrel and inertia body - damper. The principal properties of these components (weight, fastening rigidity or damping coefficient) have a significant influence on the process of selecting the cutting conditions of the drilling tool. The theoretical basis without further research is described in the aforementioned scientific studies [41,42,43]. They did not specify model parameters for vibration amplitude during the research. In the following discussion, we present a simulation of a mandrel with a dynamic vibration damper with concrete results.”
Point 9: Conclusion: it is not clear why and how this sudy resulted in the mentioned conclusions (e.g. how can you proof conclusion (6) with the conducted study??)
Response 9: Thank you for your feedback. We agree with the opponent that the article does not address the primary research of the achieved roughness of the machined surface nor the accuracy of the drilled hole dimensions. Therefore, point 6 has been deleted.
Round 2
Reviewer 1 Report
The authors have answered almost questions and there are still two need to be solved before pulication.
Response 4: We would like to provide the following explanation to this comment: Perhaps pic. 2 is not quite correct. The mathematical model (1) and the structural model (Fig. 3) describe these two elements in parallel connection.
If 'Fig. 2 is not quite correct', please provide a correct one. Furthermore, the Fig. 3 (transformation model) well matches Eq. (10) which is from Eq. (1). Eq. (1) is built according to Fig. (2), I still do not know how can Eq. (1) and Fig. 3 present the parallel relationship?
Response 10: Very good question. There are technological cases where the drilling tool rotates – e.g. classic drilling on a drilling machine, or the drilling tool does not rotate – e.g. the drilling on a turning machine. Then consider turning boring in which the cutter does not really rotate.
Rotation here means relative rotation. Either the workpiece or the boring rotates, they have relative rotation. This relative rotation brings the bending and moment. Also, the direction of the force on the cutter may not in radial or axial direction when feeding the mandrel/workpiece. Is the DOF (degree of freedom) in your model too small? Is your model accurate enough? Please give evidence of accuracy.
Author Response
Response to Reviewer 1 Comments (Round 2)
Response 4: We would like to provide the following explanation to this comment: Perhaps pic. 2 is not quite correct. The mathematical model (1) and structural model (Fig. 3) describe these two elements in parallel connection.
Point 4 (Round 2): If ‘Fig. 2 in not correct’, please provide a correct one. Furthermore, the Fig. 3 (transformation model) well matches Eq. (10) which is from Eq. (1). Eq. (1) is built according to Fig. (2), I still do not know how can Eq. (1) and Fig. 3 present the parallel relationship?
Response 4 (Round 2):
Yes, we now realize that we are inconsistent with the response we have provided. We agree that the elements of systems of the deep drilling process with the damper (shown in Figure 1) are in parallel relationship. The equation system (1) and Figure 3 also represent a parallel relationship. Then also the scheme in Figure 2 should be in parallel relationship.
We would like to remove this inconsistency and insert a new scheme in the text representing a parallel relationship. We also thank you for pointing out and pointing out this discrepancy.
At the same time, we added the following text to Line 136-137: "Figure 2 presents the parallel effect of masses m2 and m3 on mass m1."
Response 10: Very good question. These are technological cases where the drilling tool rotates – e.g. classic drilling on a drilling machine, or the drilling tool does not rotate – e.g. the drilling on a turning machine. The consider turning boring in which the cutter dos not really rotate.
Point 10 (Round 2): Rotation here means relative rotation. Either the workpiece or the boring rotates, they have relative rotation. This relative rotation brings the bending and moment. Also, the direction of the force on the cutter may not in radial or axial direction when feeding the mandrel/workpiece. Is the DOF (degree of freedom) in your model too small? Is your model accurate enough? Please give evidence of accuracy.
Response 10 (Round 2):
Yes, a very good question. We will try to give you our explanation. Of course, there are moment, bend and torsion in the system. And one can propose an equivalent model, where bending moment is equivalent to linear force, moments of inertia are equivalent to masses, angular displacements of bending are equivalent to linear displacements. But at the same time, the structural scheme (Fig. 3) and system (10) will remain the same. Of course, the real system is multidimensional and non-linear. The proposed model is one-dimensional and linear, which greatly simplifies the mathematical description.
How this simplification affects accuracy is the subject of our next prospective research.
